# The Cascade of Care for Hepatitis C Treatment in Rwanda: A Retrospective Cohort Study of the 2017–2019 Mass Screening and Treatment Campaign

**DOI:** 10.3390/v15030661

**Published:** 2023-02-28

**Authors:** Marie Paul Nisingizwe, Jean Damascene Makuza, Naveed Z. Janjua, Nick Bansback, Bethany Hedt-Gauthier, Janvier Serumondo, Eric Remera, Michael R. Law

**Affiliations:** 1Centre for Health Services and Policy Research, The University of British Columbia, Vancouver, BC V6T 1Z3, Canada; 2School of Population and Public Health, Faculty of Medicine, The University of British Columbia, Vancouver, BC V6T 1Z3, Canada; 3Rwanda Biomedical Centre, Kigali 7162, Rwanda; 4British Columbia, Center for Disease Control, Vancouver, BC V5Z 4R4, Canada; 5Harvard Medical School, Global Health and Social Medicine, Boston, MA 02215, USA

**Keywords:** hepatitis C, DAAs, direct-acting antivirals, cascade of care, mass screening and treatment, sustained virologic response, dropout, access to care, treatment initiation

## Abstract

Access to hepatitis C (HCV) testing and treatment is still limited globally. To address this, the Government of Rwanda launched a voluntary mass screening and treatment campaign in 2017. We studied the progression of patients through the cascade of HCV care during this campaign. We conducted a retrospective cohort study and included all patients screened at 46 hospitals between April 2017 and October 2019. We used hierarchical logistic regression to assess factors associated with HCV positivity, gaps in care, and treatment failure. A total of 860,801 people attended the mass screening during the study period. Some 5.7% tested positive for anti-HCV, and 2.9% were confirmed positive. Of those who were confirmed positive, 52% initiated treatment, and 72% of those initiated treatment, completed treatment and returned for assessment 12 weeks afterward. The cure rate was 88%. HCV positivity was associated with age, socio-economic status, sex, marital status, and HIV coinfection. Treatment failure was associated with cirrhosis, baseline viral load, and a family history of HCV. Our results suggest that future HCV screening and testing interventions in Rwanda and other similar settings should target high-risk groups. High dropout rates suggest that more effort should be put into patient follow-up to increase adherence to care.

## 1. Background

Hepatitis C virus (HCV) is associated with substantial morbidity and mortality worldwide, which is mainly attributable to the sequelae of infection, including liver cirrhosis, liver failure, and hepatocellular carcinoma (HCC) [1,2,3,4]. In 2019, 58 million people globally were living with chronic HCV, and approximately 290,000 people died from the virus [5]. Between 55% and 85% of acute infections progress to chronic HCV if not treated early [6]. Notably, while deaths from AIDS, malaria, and tuberculosis have declined [7], HCV-related mortality remains high, based on 2021 World Health Organization (WHO) estimates [5]. HCV is also associated with a high economic burden, both with high drug costs and the costs associated with disease progression [4,7,8,9,10]. Liver cirrhosis was among the top ten causes of death in low-income countries in 2019 [7].

The advent of a short course, well tolerated, and highly effective HCV treatment has revolutionized the management of the disease. However, while timely diagnosis and delivery of antiviral therapy can cure and prevent progression to later stages of disease, access to HCV diagnosis and treatment has been limited [9,11,12,13]. In 2019, only 21% of people living with HCV infection globally knew their status [6], and approximately 62% among those diagnosed had access to treatment. Access to HCV diagnosis and treatment is particularly challenging in sub-Saharan Africa (SSA) [14], despite the region accounting for 20% of global infections [15].

One reason why newer treatments for HCV have had limited use in SSA countries is their high cost and the costs associated with viral testing. However, in recent years, manufacturers have shown a willingness to lower these prices in resource-limited settings [16]. For example, in 2017, the government of Rwanda negotiated a price reduction for direct-acting antivirals (DAAs) that reduced treatment costs in the country from USD 1200 in 2015 to USD 60 in 2018 [17,18,19].

In 2016, the World Health Assembly set an ambitious goal of eliminating HCV by 2030. In 2017, the World Health Organization published five strategies to guide countries willing to eliminate HCV as a public health concern. These strategies include continuous collection and analysis of HCV-related data to inform policymakers, increasing screening and treatment coverage, sustainable and scalable financing, and developing new affordable diagnostics, vaccines, and treatments. In line with this, the government of Rwanda initiated a mass screening and treatment program across the country and launched a 5-year HCV elimination plan [20,21]. While similar programs have been considered elsewhere, there is a lack of evidence regarding the impact of such programs in terms of the progression of patients through the care cascade and outcomes, particularly in SSA. The HCV cascade of care is an approach used to evaluate patient retention throughout the different stages of care that are needed to achieve HCV treatment success [22].

Evidence related to the HCV cascade remains limited to high-income countries where medicine accessibility, laboratory equipment, and HCV awareness are high. A recent global systematic review that evaluated estimates on the cascade of care for HCV included mostly high-income countries, and no country in SSA had estimates of their HCV cascade [23]. Furthermore, while the number of studies reporting HCV epidemiology has been increasing, there remains limited evidence on the total number of people who initiate treatment and their treatment outcomes in SSA [3,24]. SSA is among the regions with the highest burden of HCV and other co-infections, such as HIV; therefore, it is crucial to review the HCV cascade in these settings. 

Prior studies have estimated the HCV population prevalence in Rwanda to be between 4% and 8% [19,23]. One study that assessed the cascade of care in two rural districts of Rwanda found that 83.4% of patients tested for HCV had detectable viral loads. Almost everyone considered for treatment initiated treatment, while 93.7% of patients achieved SVR12 [25]. Although this study provided information on patients’ journeys in HCV care, the findings cannot be generalized, since the study was conducted in only two of the poorest districts of Rwanda and used a small sample size. Additionally, over 40% of the patients who initiated treatment did not have SVR12 test results. Therefore, although DAA treatments are highly effective in curing HCV, treatment success represents only a small proportion of the study population. Lastly, the study did not assess factors contributing to outcomes and dropouts at each stage of care. Therefore, we studied the progression of patients through the cascade of HCV care in Rwanda’s general population using data from all public hospitals providing HCV care. We also assessed factors contributing to outcomes at each stage of the care cascade and dropouts.

## 2. Materials and Methods

### 2.1. Study Context

Over the past 20 years, Rwanda has shown progress in improving access to healthcare services across the health sector [26]. However, this same progress has not been seen in HCV care: in 2015, less than 1% of the estimated 55,000 HCV patients received drug treatment [14]. Before 2015, patients who obtained drug treatment could only do so through four hospitals that had hepatitis C specialists, and it was delivered to only a small group of comparatively wealthier patients [27].

To take advantage of the negotiated price reduction, the Rwanda Ministry of Health (MoH) launched the first voluntary mass screening and treatment campaign for HCV. The program first targeted people living with HIV and was subsequently expanded to the prison population and then to the general population. The general population screening was performed in age cohorts, starting with those who were 45 years and above and then everyone who was 15 years and older at the time of the campaign. Testing and treatment were available at one or more public health facilities (health centers or hospitals) in each of Rwanda’s 30 districts.

### 2.2. Study Design

We conducted a retrospective cohort study using secondary data to describe the HCV treatment cascade of care and assessed factors associated with outcomes and dropouts at different stages of care.

### 2.3. Study Sites and Population

We studied all patients screened at 46 of 47 public hospitals that provided HCV testing and treatment between April 2017 and October 2019. The excluded hospital is a neuropsychiatric hospital that did not provide HCV services. We also excluded patients who had indeterminate screening tests, refugees, those who died during follow-up, and those who received care from private health facilities. Eighty-five percent of Rwanda’s population seeks care at public health facilities [28], and during the mass screening, patients were only referred to public hospitals.

### 2.4. Data Sources

We used an electronic database compiled throughout the screening activities by the Rwanda Biomedical Centre (RBC). This database contained patients’ sociodemographic characteristics and other risk factors collected at the screening stage. This database has previously been used in analyses of HCV, HIV, and HBV in Rwanda [24,28,29]. We linked this database to additional information on treatment initiation and treatment outcomes extracted from patient charts. Each HCV patient had a record at the hospital that contained demographic characteristics, clinical variables (e.g., treatment and testing results, medical history, and behavior characteristics), and hospital visit dates. We used a unique identification number and demographic characteristics to link data extracted from patients to the electronic database. Data extraction was conducted by two trained nurses in each hospital who were overseen by an HCV nurse mentor.

### 2.5. Cascade of Care

The cascade of care for patients in Rwanda consisted of three major stages:

#### 2.5.1. Screening

Anti-HCV screening took place in public venues, including stadiums and playgrounds, or near other public places, such as markets, and was free of charge. Each district had at least one screening center. Trained nurses collected demographic information and blood samples from attendees at the screening venues [29,30]. The venous blood (5 mL collected in an ethylenediamine tetraacetic acid tube) samples and laboratory request forms were transported to the nearest of 13 facilities that provided enzyme-linked immune sorbent assay (ELISA) services. Murex-ELISA (version 4.0; DiaSorin S.p.A., Italy) was used across all testing sites. The samples were stored at room temperature (20–25 °C) for less than 12 h without refrigeration and then transported to laboratory facilities for ELISA testing. The sample transportation used a prespecified route to ensure efficient distribution across testing sites. At the testing sites, samples were tested for the presence of HCV antibody (anti-HCV), hepatitis B (HBsAg), and HIV. Starting in April 2019, capillary blood was drawn to test HCV antibodies using SD Bioline™ rapid diagnostic tests (RDTs) at the screening venues. Finger pricks were conducted for the present patients and immediately tested at the screening sites. These tests had near 100% sensitivity and a specificity of 99.4% [31]. Although RDTs were pre-approved by WHO for HCV screening, the Rwandan Ministry of Health, through the national referral laboratory, also conducted an internal validation to evaluate their performance before official use in Rwanda. The findings showed 97% sensitivity and 99% specificity (unpublished internal report). Patients who had a positive antibody test were provided with information on RNA viral load testing and were referred to a specialist for HCV RNA testing at their nearest hospital, where a blood sample for HCV RNA testing was obtained. From 2019, after a decentralization and task shifting intervention, patients were treated at the nearest health center by trained nurses and doctors. Those who screened as anti-HCV positive had to pay the fees associated with HCV RNA testing and additional assessment tests using health insurance or pay out-of-pocket. The cost of screening in Rwanda is approximately USD 1, while HCV RNA costs USD 9, which is equivalent to 10,000 Rwandan francs.

#### 2.5.2. HCV RNA Testing

Following referral and sample collection, trained laboratory technicians conducted HCV RNA testing at 9 HCV viral load testing sites [21]. Some blood samples were collected at hospitals that did not provide viral load testing, so they were transferred on the same day to testing sites. Each sample had patient and hospital identification information. The hospital at which the sample was taken called patients to inform them of their results. Patients who were confirmed to be positive received counseling and were provided information on the treatment process by an HCV specialist and referred for treatment. Figure 1 provides more details on the screening and testing process.

#### 2.5.3. Treatment

All patients who had a positive HCV RNA test (detectable 15–20 copies/mL) received treatment free of out-of-pocket charges [29,30]. During the study period, the HCV treatment available in Rwanda included ledipasvir/sofosbuvir (Harvoni^®^) and sofosbuvir/daclatasvir (DCV). DAAs are known to have a higher cure rate (90%) and fewer side effects than other therapeutic options [23]. After treatment completion, all patients were asked to return to the hospital for viral load testing. All patients whose HCV results were undetectable (below 15–20 copies/mL) within 12 weeks after the date of treatment completion were considered cured.

### 2.6. Outcome Variables

Table 1 lists the outcomes that we used to describe the cascade of care for patients with HCV. Along with the number and proportion of patients who completed each stage of care, we also estimated the proportion of patients who dropped out at two stages of the cascade: (1) patients with no confirmed diagnosis and (2) patients who did not return for their final check-up 12–24 weeks following treatment completion. We defined treatment failure as having an HCV viral load that was above 20 copies/mL between 12 and 24 weeks post-treatment.

### 2.7. Other Variables

We included covariates identified in previous studies [14,22,32], including the following demographic and socio-economic factors: age, sex, Social Economic Status, health insurance, and marital status. In Rwanda, households are classified into four different Social Economic Status (SES) (known as ubudehe) categories based on their household income and assets, with category 1 being the lowest wealth group and category 4 representing the highest wealth group [33]. This variable includes a category called unknown that represents new families that have not yet been classified into any SES category.

We also collected clinical and behavioral features, including previous HCV treatment, hepatitis B vaccination status, previous transfusion and surgical history, family history of HCV, comorbidities (hepatitis B, HIV, diabetes, liver diseases, cancer, and renal failure status), and traditional operation practices. For treatment outcomes, we included variables related to the type of drugs a patient received, previous HCV treatment history, baseline viral load, and cirrhosis status (stage of HCV). Lastly, we included hospital-level variables, such as number of staff and hospital type (referral, teaching, provincial, or district).

### 2.8. Data Analysis

We started by generating descriptive statistics to estimate the proportion of patients at each stage of care and treatment success, followed by bivariate analysis using chi-square tests and t-tests to assess the association between covariates and outcomes. Following this, we used a hierarchical logistic regression model to assess factors associated with HCV positivity, dropout at different stages of care, and treatment failure while controlling clustering in the data. Continuous variables were centered or standardized for modeling and interpretation purposes. Some covariates had missing values, and we included a missing category for each variable with missing values to enable us to retain all observations in the analysis [34].

Our model-building process included three steps. First, we estimated the interclass correlation coefficient using an empty model. Second, we included all patient-level and hospital-level variables that were significant in the bivariate analysis and those that were important based on prior literature. Covariates were included in the model using a forward stepwise selection process using the AIC and the likelihood ratio test. Finally, we tested all interaction terms between variables for significance at α = 0.05 and retained those that were significant. All analyses were conducted using STATA 16.0.

## 3. Results

### 3.1. Descriptive Characteristics

The characteristics of our cohort are shown in Table 2. Our cohort included 860,801 people who were screened for HCV at 46 different public hospitals. The average age was 42 years old (Standard Deviation (SD): 18), and overall, the participants ages ranged from 15 to 89 years old. Approximately half were in the third category of SES (47.7%), 68% were female, and 63.6% were married or in a union. Approximately 18.8% did not have health insurance. In terms of comorbidities, 2.9% were HIV positive, and 3.0% had hepatitis B (HBsAg). In terms of previous medical history, 1.7% had previously been diagnosed with liver disease, and 2.8% had a family history of HCV.

### 3.2. Cascade of Care

The overall progression through each stage of the cascade of care is shown in Figure 2. Below, we will discuss the results from each individual step in the cascade:

### 3.3. Anti-HCV Seroprevalence

As shown in Figure 2, a total of 48,680 people, or 5.7%, had a positive anti-HCV test. The odds of being anti-HCV positive increased with age. Other factors associated with being positive included being HIV positive (OR: 1.91, 95% CI: 1.80 to 2.03), having a family history of viral hepatitis (OR: 1.1, 95% CI: 1.04 to 1.18), and having had a traditional operation (OR: 1.31, 95% CI: 1.27 to 1.36). Factors associated with lower positivity rates included being HBV positive (OR: 0.72, 95% CI: 0.67 to 0.77) and being vaccinated for HBV (OR: 0.53, 95% CI: 0.51 to 0.55). More details are provided in Table 3.

### 3.4. HCV RNA Testing

Of those who screened positive, 43,893 (90%) completed an HCV RNA test. Of these individuals, 25,031 (57%) tested positive. This suggests a chronic HCV infection prevalence of 2.9% among individuals who were screened during our study period and 3.2% among those who were HCV RNA tested. The socio-economic and demographic characteristics associated with a positive HCV RNA test included age, SES, marital status, and sex. HIV positivity (OR: 2.51, 95% CI: 2.20 to 2.87) and having hypertension (OR: 2.05, 95% CI: 1.80 to 2.34) were associated with a positive HCV RNA test. Factors associated with lower rates of HCV positivity included being HBV positive, having ever been medically operated upon, and not being vaccinated for HBV (Table 4).

### 3.5. Dropout at HCV RNA Testing 

Of the 48,680 individuals who screened anti-HCV positive, 10% (4787) did not return for HCV RNA testing (Figure 2). Table 2 shows the descriptive characteristics of those who were lost to follow-up at this stage. Our model found that patients who were 44–54 years old (OR: 0.58, 95% CI: 0.46 to 0.74) and 55–64 years old (OR: 0.4, 95% CI: 0.35 to 0.47) were less likely to drop out compared to those who were younger than 44 years old, while those who were above 64 years old had four-fold higher odds (OR: 4.54, 95% CI: 4.03 to 5.12). Other factors associated with lower dropout rates included being male, being HIV positive, being diabetic, having a family history of viral hepatitis, and being in a higher SES category. Factors related to higher dropout rates included being married or in a union, having community insurance, and not being vaccinated for HBV (Table 5).

### 3.6. Treatment Success (Achieving SVR)

Of the 25,031 people who were HCV RNA positive, 12,940 (52%) initiated treatment, as recorded in hospital registers. Of these, 9332 (72%) completed treatment and returned for SVR assessment. Our data suggested that 0.4% died in hospital before treatment completion, and 27.6% initiated treatment but did not return for their final assessment. The majority (80.4%) of people who died were above 64 years old. Of those who completed treatment and returned for SVR assessment, 8232 (88%) were cured (Figure 2). Of those who failed treatment, approximately half were above 64 years old, and 59% were female (Appendix A in the Appendix A). As shown in Table 6, treatment failure was associated with increased baseline viral load (OR: 1.14, 95% CI: 1.02 to 1.28), having cirrhosis (OR: 1.71, 95% CI: 1.20 to 2.43), having had previous treatment for HCV (OR: 1.9, 95% CI: 1.34 to 2.70), and having a family history of HCV (OR: 1.57, 95% CI: 1.10 to 2.25).

### 3.7. Dropout at Final SVR Test

Of the 3552 individuals who did not complete an SVR assessment, most were above 64 years old (N = 2273, 64%) and female, and the majority did not have comorbidities or a prior history of HCV. More details on descriptive statistics can be found in Appendix A in the Appendix A. In our final model (Table 6), patients who were more than 64 years old (OR: 1.42, 95% CI: 1.17 to 1.72) were less likely to complete their SVR assessment. In contrast, males and patients with comorbidities, including HIV, HBV, and cirrhosis, were more likely to attend their final checkups. Finally, patients with a family history of viral hepatitis (OR: 0.14, 95% CI: 0.08 to 0.24), those who had ever been medically operated on (OR: 0.39, 95% CI: 0.27 to 0.56), and those who were fully or partially vaccinated (OR: 0.14, 95% CI: 0.11 to 0.16) were more likely to undergo their SVR test.

## 4. Discussion

HCV remains a significant source of morbidity and mortality in low-and middle-income countries (LMICs). The advent of lower prices for both DAA therapy and HCV testing has increased the opportunities for the treatment of a broader proportion of the population. Our study of the first 2 years of voluntary mass screening for HCV in Rwanda found that a notable proportion of the population was tested, and 2.9% of the total screened population was confirmed chronic HCV-positive. People who attended the screening campaign were, on average, older than the general population [35] but had HIV and HBV prevalences that were comparable to the prior general population estimates [36]. Of the nearly 13,000 individuals for whom we had records, HCV was cured 88% of the time. However, we also found a significant number of dropouts at each stage of the HCV care cascade, which merits attention in the design of future interventions.

### 4.1. Anti-HCV Screening and HCV RNA Testing

The observed HCV prevalence and viremia rate is consistent with the existing literature [30,36,37]. The likelihood of HCV positivity increased with age and was inversely related to SES. As previously reported in other studies [23,38], HCV increases with age, mainly due to a history of unsafe medical procedures and unsterile injection practices in SSA [14,31,39,40]. Our findings are consistent with prior studies in Rwanda [29,38], Tanzania [41], and Egypt [42,43] that showed an association between HCV diagnosis and being economically disadvantaged due to low access to preventive and treatment services. As reported in other studies, we also found that chronic HCV is more common in men than in women [44].

HIV positivity and hypertension were associated with the presence of HCV. In our study, 7.2% of HIV patients had a positive HCV RNA test. A recent global meta-analysis reported that the odds of HCV were six-fold higher among HIV-positive patients compared to those who were HIV negative [45]. As both infections have common routes of transmission, including needle sharing and contaminated blood products, the high coinfection rate is not surprising [46]. In contrast, consistent with prior studies conducted in Rwanda and Morocco [39,47], those with HBV coinfection were less likely to be confirmed HCV positive.

### 4.2. Treatment Initiation and Success

In our study, only half of all diagnosed patients initiated treatment through a public hospital. HCV treatment initiation is still a significant challenge in SSA due to the limited availability of drugs [16]. In Rwanda, healthcare providers have previously reported being out of stock of HCV medicines and testing commodities, which might have led to the low initiation rate observed [48]. Similarly, a study conducted in Tanzania among people who injected drugs reported that none of the HCV-diagnosed individuals had initiated treatment [41].

Our observed treatment success rate was similar to a prior study on the efficacy of ledipasvir/sofosbuvir, which found that 87% of patients in Rwanda achieved SVR12 [49]. Patients in both studies were treated with the first generation of DAAs. In two single-arm studies that evaluated the efficacy of the new generation of DAAs (sofosbuvir/velpatasvir and sofosbuvir/velpatasvir/voxilaprevir), treatment success was above 90% [50]. The difference in treatment success is mainly due to the use of the newer DAAs, which are known to have higher treatment success rates [50,51]. Currently, patients with advanced stages of HCV or previously failed treatment are recommended to use the new generation of DAAs [50,51]. However, the availability of these medicines is still limited in many LMICs [16], which might explain the higher rate of failure in this group. In addition, it is essential to note that although patients are given medication instructions on timing and daily intake, adherence to these instructions and medications is still unknown. Treatment failure can also be associated with non-adherence to treatment.

### 4.3. HCV RNA and SVR Testing Dropout

Similar to previous studies [24,52,53,54], our study showed significant dropout at the HCV RNA testing and SVR stages. The older group was less likely to undergo HCV RNA and SVR testing, while those in the higher SES categories were more likely to attend. Similarly, Naveed et al. [55], in their study describing the cascade of care for British Columbia, Canada, showed that the older generation had higher rates of dropout, while those with comorbidities, such as HIV and cirrhosis, were more likely to be in care.

Comorbidities (HIV, HBV, diabetes, hypertension, and cirrhosis) and a family history of viral hepatitis also contributed to patients being more likely to complete HCV RNA and SVR testing. Unlike patients with a single disease, patients with multimorbidity are more likely to seek care [56], likely in part because they are aware of early treatment benefits, are familiar with the health system, and already have established relationships with healthcare providers. Additionally, they are aware that they are at an increased risk of death [57] if not treated. Lastly, it should be noted that patients received a complete regimen at once and that patients without a prior medical history are less likely to return for care [58].

We also found that those with community insurance were more likely to drop out. It is important to note that although HCV medicines were free during the mass campaign, patients had to pay the costs associated with assessment tests. The HCV RNA test costs USD 9, which is equivalent to approximately 10,000 Rwandan francs [49]. Patients who have insurance pay 10–15% of the total cost. However, the country still has approximately 40% of the population living under the poverty line [59], and this cost may be too high in many cases. Additionally, unlike other primary care services provided at the closest health centers, when the HCV mass campaign started, patients who screened positive were referred to hospitals for HCV RNA testing and had to return for a final viral load checkup after treatment completion. The burden associated with travel cost and distance might also have contributed to these gaps in service delivery. There was also low awareness and knowledge of HCV in the general population [60], which could be addressed through continued public education regarding HCV. Recently, the government has decentralized all HCV services to the low level of health facility (health centers) [21], which is expected to improve public awareness and testing and treatment uptake. The HCV program has been integrated into the existing healthcare system. Further analysis assessing the impact of this change on the cascade of care and dropout estimates would be helpful.

### 4.4. Limitations

Our study has some limitations that are worth noting. First, it is important to note that the mass screening and treatment campaign was voluntary in nature. Therefore, the characteristics of the people who participated may differ from those who did not. While this might limit the generalizability of some findings, particularly the population rate of HCV infection, it would not impact the outcomes of the cascade of care among participants at the later stages. Second, our data were limited to what was available in patient records, which means that some important factors, such as distance to a health facility, use of injectable drugs, and sexual behaviors, could not be included. The patients’ records also only included deaths that took place in the hospital. Thus, patients who died in the community might have been incorrectly misclassified as lost to follow-up. We believe the bias associated with this misclassification would likely be small because of the relatively short follow-up time involved. In addition, the treatment initiation rate could only be calculated for those who had records in hospital registers. However, given the high cost of HCV treatment and the availability of free, publicly financed medicines in Rwanda, we are confident that the number we missed would be small. Finally, we had missing data for many of our variables of interest. However, we used a missing indicator approach [34] in our analysis for each variable to utilize most of the sample in the analysis. In addition, both our reported missing indicator and complete case analyses resulted in similar findings.

## 5. Conclusions

We found that the mass screening campaign reached many people, especially those with a prior history of viral hepatitis and other comorbidities. Our findings suggest that future HCV screening and testing interventions in Rwanda and other countries should focus on the older population, those with lower SES levels, and individuals with comorbidities. We also observed that the success rate was lower than that of patients treated with the new generation of DAAs, and we recommend further negotiation for the availability of newer combination DAAs, especially for cirrhotic patients and those who previously failed HCV treatment. Finally, efforts should be made to increase adherence to SVR assessment and the completion of all stages of care.

## Figures and Tables

**Figure 1 viruses-15-00661-f001:**
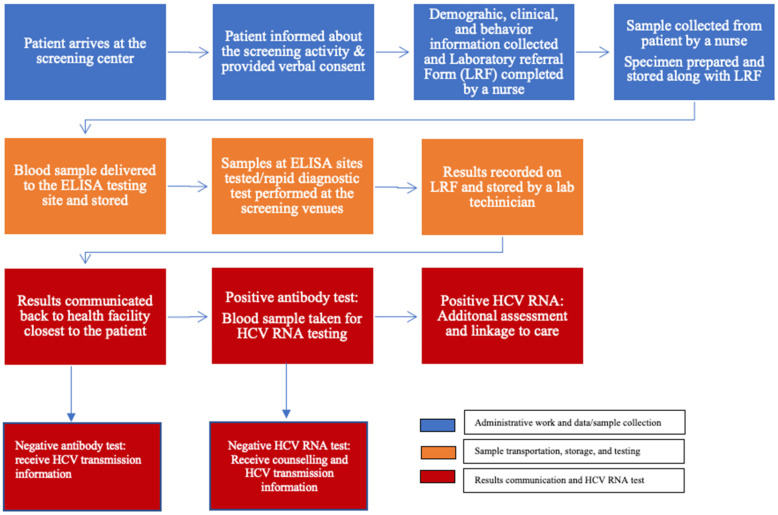
Mass screening and testing process.

**Figure 2 viruses-15-00661-f002:**
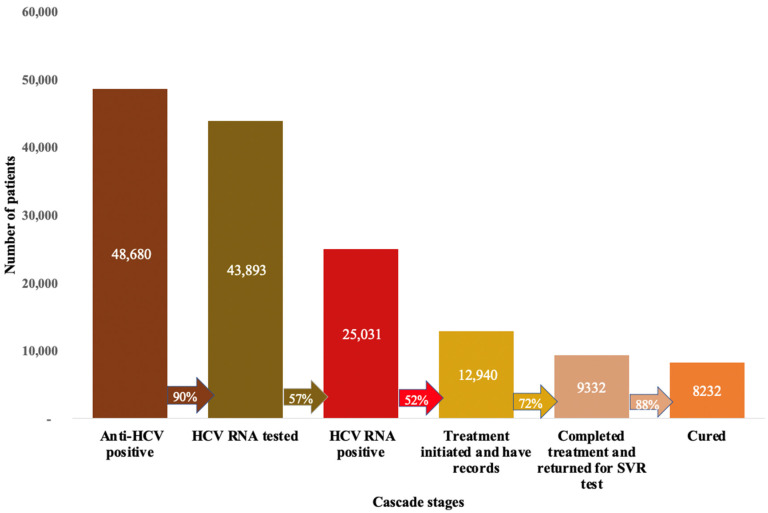
HCV cascade of care in Rwanda for the 2017–2019 mass screening and treatment campaign cohort.

**Table 1 viruses-15-00661-t001:** Stages in the HCV cascade of care.

Outcome	Numerator	Denominator
Proportion of patients who were anti-HCV positive	Number of patients with a positive HCV antibody test	Number of patients screened
HCV RNA viremia rate	Number of patients who tested HCV RNA positive	Number of patients who screened positive for anti-HCV with HCV RNA test conducted
Proportion of patients who dropped out of care before HCV RNA viremia testing	Number of patients who screened positive for anti-HCV and did not return for an HCV RNA test	Number of patients who screened positive for anti-HCV
Proportion of patients who initiated treatment	Number of patients who initiated treatment with available records in hospital registers	Number of patients who were confirmed HCV RNA positive
Proportion of patients who achieved SVR (treatment success)	Number of patients whose HCV viral load result was below 20 copies/mL within 12–24 weeks post-treatment and returned for SVR test	Number of patients who completed treatment with available records and returned for SVR assessment (excluding those who died)
Proportion of patients who did not return for SVR tests	Number of patients who initiated treatment and did not come back for SVR assessment	Number of patients who initiated treatment with available records in the registers (excluding those who died)

**Table 2 viruses-15-00661-t002:** Characteristics of patients screened and treated during the mass screening campaign for hepatitis C virus in Rwanda (2017–2019).

	Total Sample		Anti-HCV Positive		HCV RNA Testing			
Variables	Total Sample	Total (%)	Total Number Screened Anti-HCV Positive	Anti-HCV Positive (%)	Total Number HCV RNA Tested	HCV RNA Positive (%)	Total Dropouts	Dropout (%)
Age								
<44	426,186	59.4	9378	19.4	3324	13.3	686	15.1
44–54	112,264	15.6	6165	12.8	2977	11.9	265	5.8
55–64	100,439	14	9918	20.6	5498	22	340	7.5
64+	78,493	10.9	22,848	47.3	13,203	52.8	3252	71.6
Sex								
Female	555,389	68.4	32,159	66.5	16,442	65.7	3259	68.1
Male	256,701	31.6	16,580	33.5	8587	34.3	1528	31.9
SES categories								
Category 1	104,681	12.2	8077	23.3	4971	19.9	1163	25.5
Category 2	260,537	30.3	12,211	35.2	6854	27.4	1510	33.1
Category 3 or 4 or unknown	495,583	57.6	14,456	41.6	8201	52.7	1888	41.3
Marital status								
Single	109,552	15.1	1625	4.9	910	4.8	126	4.6
Married or in union	461,231	63.6	21,522	64.9	11,911	63.4	1707	62
Separated/divorced/widowed	153,919	21.2	9927	30.2	5972	31.8	920	33.4
Health insurance								
No insurance	161,853	18.8	7867	15.9	172	19.3	1045	21.8
Community insurance	657,067	76.3	40,076	81.1	19,517	78	3643	76.1
Private or other government insurance (RAMA/RSSB)	41,881	4.9	1437	3	694	2.8	99	2.1
Diabetes status								
No	694,243	99	32,956	98	18,681	97.5	2835	98.9
Yes	7353	1	676	2	475	2.5	32	1.1
HTA status								
No	448,936	97.7	30,204	92.5	17,626	90.2	1845	92.4
Yes	10,653	2.3	2455	7.5	1915	9.8	152	7.6
Renal failure status								
No	453,150	98.8	30,926	98.1	18,461	98.7	2078	97.9
Yes	5365	1.2	592	1.9	247	1.3	44	2.1
HIV status								
Negative	680,082	97.1	31,789	94.6	17,719	92.8	2836	98.6
Positive	20,071	2.9	1806	5.4	1371	7.2	40	1.4
HBV result								
Negative	829,644	97	42,548	97.7	20,278	98.7	4323	97.6
Positive	25,652	3	984	2.3	267	1.3	105	2.4
Family History of viral hepatitis C								
No	837,090	97.2	32,317	95.4	17,962	96	4709	98.4
Yes	23,711	2.8	1543	4.6	1006	4	78	1.6
Ever been traditionally operated on								
No	597,913	85.1	26,955	79.3	15,598	80.4	2312	81.7
Yes	104,688	14.9	7020	20.7	3801	19.6	517	18.3
Ever been transfused								
No	683,769	97.3	32,931	96.9	18,770	96.8	2759	97.5
Yes	18,729	2.7	1047	3.1	617	3.2	70	2.5
Ever been medically operated on								
No	665,094	94.7	32,108	94.3	18,430	94.7	2674	94.6
Yes	37,518	5.3	1945	5.7	1022	5.3	152	5.4
Ever been diagnosed with liver disease								
No	399,965	98.3	18,211	97.58	15,000	97.9	895	98.2
Yes	6947	1.7	452	2.42	310	2.1	16	1.8
HBV vaccination								
Partially or fully vaccinated	53,723	6.24	5245	31.3	4519	32.8	66	1.4
Not vaccinated	807,078	93.76	43,435	68.7	20,512	67.2	4721	98.6

**Table 3 viruses-15-00661-t003:** Unadjusted and adjusted associations between anti-HCV seroprevalence and socio-economic, demographic, clinical, and behavior factors.

		Unadjusted Model		Adjusted Model ^†^	
Variables	N	OR	95% CI	OR	95% CI
Age					
<44	426,186	1	1.00, 1.00	1	1.00, 1.00
44–54	112,264	2.53	2.44, 2.61	2.41	2.33, 2.50
55–64	100,439	4.66	4.52, 4.80	4.45	4.31, 4.60
64+	78,493	18.1	17.64, 18.61	15.1	14.67, 15.59
Missing	143,419	0.064	0.06, 0.07	0.0093	0.01, 0.01
SES					
Category 1	104,681	1	1.00, 1.00	1	1.00, 1.00
Category 2	260,537	0.66	0.64, 0.68	0.84	0.81, 0.87
Category 3 or 4 or unknown	336,580	0.59	0.58, 0.61		
Missing	159,003	1.45	1.40, 1.50		
Marital status					
Single	109,552	1	1.00, 1.00	1	1.00, 1.00
Married or in union	461,231	2.9	2.75, 3.06	1.47	1.39, 1.55
Separated/divorced/widowed	153,919	3.59	3.39, 3.79	1.25	1.18, 1.33
Missing	136,099	6.69	6.33, 7.07	6.5	5.95, 7.11
HIV					
No	680,082	1	1.00, 1.00	1	1.00, 1.00
Yes	20,071	2.2	2.09, 2.32	1.91	1.80, 2.03
Missing	160,648	1.55	1.51, 1.59	0.91	0.76, 1.08
Family history of viral hepatitis					
No	677,376	1	1.00, 1.00	1	1.00, 1.00
Yes	23,711	1.27	1.21, 1.35	1.1	1.04, 1.18
Missing	159,714	1.48	1.44, 1.52	1.51	1.20, 1.90
Ever been traditionally operated on					
No	597,913	1	1.00, 1.00	1	1.00, 1.00
Yes	104,688	1.52	1.48, 1.56	1.31	1.27, 1.36
Missing	158,200	1.55	1.51, 1.59	0.23	0.17, 0.32

^†^ Controlled for sex, diabetes status, transfusion and surgical history, hepatitis B status, and HBV vaccination status.

**Table 4 viruses-15-00661-t004:** Unadjusted and adjusted associations between HCV RNA positivity and socio-economic, demographic, clinical, behavioral, and hospital characteristics factors.

		Unadjusted Model		Adjusted Model ^†^	
Variables	N	OR	95% CI	OR	95% CI
Age					
<44	8803	1	1.00, 1.00	1	1.00, 1.00
44–54	5991	1.5	1.36, 1.58	1.65	1.51, 1.80
55–64	9744	1.8	1.69, 1.93	1.89	1.75, 2.04
>64	19,164	2.7	2.51, 2.82	2.88	2.67, 3.10
Missing	191	0.2	0.11, 0.25	0.33	0.20, 0.53
Social Economic Status					
Category 1	7061	1	1.00, 1.00	1	1.00, 1.00
Category 2	10,957	0.7	0.67, 0.77	0.8	0.74, 0.87
Category 3 or 4 or unknown	12,839	0.7	0.66, 0.76	0.78	0.72, 0.85
Missing	13,036	0.3	0.32, 0.36	0.3	0.27, 0.33
Sex					
Female	28,834	1	1.00, 1.00	1	1.00, 1.00
Male	15,059	1.1	1.04, 1.13	1.34	1.27, 1.42
Marital status					
Single	1528	1	1.00, 1.00	1	1.00, 1.00
Married or in union	20,191	1.3	1.12, 1.41	0.85	0.75, 0.97
Separated/divorced/widowed	9292	1.5	1.30, 1.66	0.84	0.73, 0.97
Missing	12,882	0.7	0.59, 0.75	1.24	1.04, 1.49
Hepatitis B Ag results					
Negative	38,922	1	1.00, 1.00	1	1.00, 1.00
Positive	880	0.5	0.39, 0.55	0.53	0.44, 0.65
Missing	4091	15	13.29, 16.60	109.4	95.49, 125.34
HIV					
No	29,622	1	1.00, 1.00	1	1.00, 1.00
Yes	1789	2	1.80, 2.31	2.51	2.20, 2.87
Missing	12,482	0.4	0.42, 0.46	0.49	0.41, 0.58
Hypertension					
No	29,048	1	1.00, 1.00	1	1.00, 1.00
Yes	2323	2.3	1.99, 2.54	2.05	1.80, 2.34
Missing	12,522	0.4	0.33, 0.37	0.12	0.11, 0.14
Ever been traditionally operated on					
No	25,212	1	1.00, 1.00	1	1.00, 1.00
Yes	6623	1	0.90, 1.02	0.9	0.84, 0.97
Missing	12,058	0.4	0.41, 0.46	0.66	0.45, 0.97
Ever been medically operated on	43,893				
No	30,066	1	1.00, 1.00	1	1.00, 1.00
Yes	1853	0.7	0.60, 0.74	0.66	0.58, 0.74
Missing	12,755	0.4	0.41, 0.45	1.67	0.98, 2.83
HBV vaccination status					
Partially or fully vaccinated	5635	1	1.00, 1.00	1	1.00, 1.00
Not vaccinated	38,258	0.3	0.29, 0.34	0.49	0.45, 0.53

^†^ Controlled for transfusion history, family history of viral hepatitis, and diabetes status.

**Table 5 viruses-15-00661-t005:** Unadjusted and adjusted associations between dropout at HCV RNA testing stage and socio-economic, demographic, clinical, behavioral, and hospital characteristics factors.

		Unadjusted Model		Adjusted Model ^†^	
Variables	N	OR	95% CI	OR	95% CI
Age					
<44	8987	1	1.00, 1.00	1	1.00, 1.00
44–54	6250	0.6	0.51, 0.69	0.5	0.42, 0.59
55–64	10,081	0.48	0.42, 0.55	0.4	0.35, 0.47
>64	22,927	2.53	2.31, 2.77	4.54	4.03, 5.12
Missing	435	14.23	10.78, 18.78	491	328.16, 734.68
Social Economic Status					
Category 1	8224	1	1.00, 1.00	1	1.00, 1.00
Category 2	12,462	0.66	0.60, 0.72	0.77	0.70, 0.86
Category 3 or 4 or unknown	14,747	0.7	0.64, 0.76	0.95	0.86, 1.05
Missing	13,247	0.066	0.06, 0.08	0.011	0.01, 0.01
Sex					
Female	32,100	1	1.00, 1.00	1	1.00, 1.00
Male	16,580	0.86	0.81, 0.92	0.87	0.80, 0.95
Health insurance					
No insurance	7867	1		1	1.00, 1.00
Community insurance	39,312	0.71	0.65, 0.76	1.7	1.40, 2.06
Private or other government insurance (RAMA/RSSB)	1501	0.41	0.32, 0.51	1.13	0.82, 1.56
HIV					
No	32,449	1	1.00, 1.00	1	1.00, 1.00
Yes	1829	0.27	0.19, 0.37	0.37	0.25, 0.54
Missing	14,402	1.34	1.25, 1.43	0.18	0.10, 0.31
Diabetes					
No	33,611	1	1.00, 1.00	1	1.00, 1.00
Yes	693	0.68	0.47, 0.99	0.57	0.43, 0.76
Missing	14,376	1.43	1.34, 1.54	1.6	1.07, 2.40
Family history of viral hepatitis C					
No	32,958	1	1.00, 1.00	1	1.00, 1.00
Yes	1583	0.57	0.45, 0.72	0.63	0.41, 0.98
Missing	14,139	1.44	1.35, 1.54	0.28	0.16, 0.50
HBV vaccination status					
Partially or fully vaccinated	5701	1	1.00, 1.00	1	1.00, 1.00
Not vaccinated for hepatitis B	42,979	6.62	5.15, 8.51	8.77	6.66, 11.54

^†^ Controlled for marital status, hepatitis B status, hypertension status, and total number of hospital staff.

**Table 6 viruses-15-00661-t006:** Adjusted and unadjusted relationships between treatment failure, dropout at SVR assessment stage, and socio-economic, demographic, clinical, behavioral, and hospital characteristics factors.

	Treatment Failure					Dropout at SVR Assessment Stage				
	N	Unadjusted model		Adjusted model ^†^			Unadjusted model		Adjusted model ^†^	
		OR	95% CI	OR	95% CI	N	OR	95% CI	OR	95% CI
Baseline viral load	9332	1.12	1.04, 1.39	1.14	1.02, 1.28	-	-	-	-	-
Age										
<44	1227	1	1.00, 1.00	1	1.00, 1.00	1591	1	1.00, 1.00	1	1.00, 1.00
44–54	1262	1.02	0.79, 1.32	1.34	0.93, 1.91	1571	1.02	0.83, 1.26	1.1	0.88, 1.39
55–64	2343	0.94	0.75, 1.18	1.18	0.86, 1.63	2968	0.96	0.80, 1.15	1.11	0.91, 1.35
64+	4500	0.94	0.76, 1.15	1.12	0.82, 1.53	6754	1.9	1.63, 2.23	1.42	1.17, 1.72
Sex										
Female	5654	1	1.00, 1.00	1	1.00, 1.00	7939	1	1.00, 1.00	1	1.00, 1.00
Male	3678	1.01	0.87, 1.16	1.1	0.90, 1.33	4945	0.81	0.73, 0.90	0.76	0.67, 0.85
SES										
Category 1	1835	1	1.00, 1.00	1	1.00, 1.00	2573	1	1.00, 1.00	1	1.00, 1.00
Category 2	2541	0.97	0.79, 1.18	0.98	0.76, 1.25	3296	0.89	0.77, 1.03	0.96	0.82, 1.14
Category 3 or 4 or unknown	3224	1.15	0.95, 1.40	1.13	0.88, 1.45	5174	0.64	0.56, 0.73	0.73	0.62, 0.84
Missing	1732	1.41	1.12, 1.78	0.61	0.43, 0.86	1841	3.58	3.01, 4.26	2.53	2.08, 3.08
HIV positive										
No	7795	1	1.00, 1.00	1	1.00, 1.00	11,075	1	1.00, 1.00	1	1.00, 1.00
Yes	922	0.8	0.62, 1.03	1.09	0.81, 1.46	1125	0.31	0.25, 0.38	0.3	0.24, 0.38
Missing	615	5.92	4.73, 7.40	0.77	0.20, 3.01	684	0.076	0.05, 0.13	0.059	0.03, 0.11
HBV Ag positive										
No	8873	1	1.00, 1.00	1	1.00, 1.00	12,291	1	1.00, 1.00	1	1.00, 1.00
Yes	140	1.27	0.79, 2.05	1.11	0.57, 2.18	274	0.86	0.57, 1.31	0.6	0.37, 0.99
Missing	319	2.8	2.00, 3.92	1.48	0.84, 2.63	319	0.73	0.54, 0.99	0.79	0.50, 1.22
Cirrhotic										
No	8898	1	1.00, 1.00	1	1.00, 1.00	12,440	1	1.00, 1.00	1	1.00, 1.00
Yes	434	1.12	0.81, 1.54	1.71	1.20, 2.43	444	0.066	0.03, 0.13	0.059	0.03, 0.12
Family history of viral hepatitis C										
No	8220	1	1.00, 1.00	1	1.00, 1.00	11,376	1	1.00, 1.00	1	1.00, 1.00
Yes	437	1.57	1.17, 2.12	1.57	1.10, 2.25	721	0.42	0.30, 0.57	0.14	0.08, 0.24
Missing	675	5.96	4.79, 7.42	3.62	1.07, 12.31	787	0.66	0.53, 0.81	0.18	0.08, 0.39
Ever been medically operated on										
No	8351	1	1.00, 1.00	-	-		1	1.00, 1.00	1	1.00, 1.00
Yes	306	1.34	0.90, 1.97	-	-		0.57	0.47, 0.68	0.39	0.27, 0.56
Missing	675	5.8	4.71, 7.29	-	-		1.21	0.56, 1.40	1.44	0.89, 2.32
HBV vaccination status										
Not vaccinated	6985		-	-	-		1	1.00, 1.00	1	1.00, 1.00
Partially or fully vaccinated	2347	-	-	-	-		0.11	0.09, 0.13	0.14	0.11, 0.16

^†^ Controlled for marital status, health insurance, hypertension, diabetes, and total number of staff.

## Data Availability

Our ethical approval from the National Rwanda Ethics Board only permitted data access by the research team.

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
