# Peer review of "The Cascade of Care for Hepatitis C Treatment in Rwanda: A Retrospective Cohort Study of the 2017–2019 Mass Screening and Treatment Campaign"

_viruses, 2023, doi:10.3390/v15030661_

Round 1

Reviewer 1 Report

Abstract:

1.     Please add “family” to the following phrase to avoid misunderstandings:

Treatment failure was associated with cirrhosis, baseline viral load, and “family” history of HCV.

Background:

2.     Regarding the following statement:  

“Notably, while deaths from AIDS, malaria and tuberculosis have declined, HCV-related mortality has continued to climb (6).”

To support the statement the authors use reference 6 (Sonderup MW et al.) but the data provided in reference 6 is from 2015 and before. Could you use a more updated reference for this statement (e.g. latest 2021 WHO report) or provide the time frame for it so readers can have an idea till when it applies?

3.     This is only a suggestion, but moving the phrase “In 2016, the World Health Assembly set an ambitious goal of eliminating HCV by 2030” to the next paragraph would make sense.

Example: “In 2016, the World Health Assembly set an ambitious goal of eliminating HCV by 2030. In 2017(2), the World Health Organization published five strategies to guide countries willing to eliminate HCV as a public health concern.”

4.     Regarding the following statement: One study that assessed the cascade of care in two rural districts of Rwanda found that 83.4% of patients tested for HCV had detectable viral loads. Almost everyone considered for treatment initiated treatment, while 93.7% of patients achieved SVR12 (23).

I would suggest the authors clarify the data reported in this phrase:

“83.4% of the patients tested HCV RDT positive and with results had detectable viral loads. Almost everyone considered for treatment (99%) initiated treatment, while 93.7% of patients with tests results achieved SVR12 (23).

Overall, 43.7% of the patients that undergo treatment had no test results for SVR12 in this study and it is relevant to explain it or at least suggest it that to the reader.

Methods:  

5.       Section 2.2

Patients are screened at 46 public hospitals. Could the authors provide the overall number of public/private existing hospitals in the country or an estimator of the % of the population of the country that the included hospitals cover?

Results:

6.       Section 3.1

Please add SD value to the following phrase: “The average age was 42(SD:)”

Also, if it would be possible it will be interesting to have the range for the variable age.

Regarding the following statement “In terms of comorbidities, 2.9% were HIV positive, 3.0% had hepatitis B(HBsAg), and 2.3% had hypertension”. I do not understand well why to remark hypertension prevalence among the other variables with more relevance for the study as transfusions, surgeries or vaccination history.

7.       General commentary about ORs calculation

Did the authors try to calculate ORs excluding missing values? Although losing observations, estimates would be more reliable in the case that missing values were distributed randomly in the sample.

Did the authors check if distribution of missing values are random or follow a pattern?

8.       Table 3

Table 3 is not reflecting missing values, while Table 4 and 5 do. Please, add missing values information to Table 3.

9.       Table 4

Is table 4 title correct? It seems to be the same than table 5. However, table 5 is referring to dropouts and table 4 to HCV RNA test positivity. Please review and adapt the table title to its content.

Table 4. OR calculated for Hepatitis B Ag results is 15 (too high) in the unadjusted model, and 109.4 in the adjusted model. Does the researchers check for interactions or confounding factors in the final model to explain these results? Could the authors suggest and explanation for these results? Which characteristics has missing group for this variable to obtain such OR values?

Also, for the variable “Ever been operated” there are two missing results. Please eliminate the one that proceed or explain the differences between both.

Discussion

10.    General comment

As a general suggestion, it would be interesting to start the discussion with a review of the general situation. The population sample was attended in 46 hospitals, that covered % population. The sample is not representative of general population because it was targeted first to HIV population, followed by prison population and after, general. However, % of general population among the sample was X and it may/may not be representative (This data is missing in the results section).

11.    Section 4.1

Please delete the remark of hypertension as a risk factor to have more HCV. Hypertension and diabetes can be a result of the infection or a result of other risk factors/intrinsic characteristics of patients. In the case of cirrhosis, it is most likely a consequence of HCV infection. Please review all comments made in the manuscript and instead of saying “hypertension or other comorbidity was found to be a risk factor for HCV” say “hypertension or other comorbidity was associated to the presence/diagnosis of HCV”.

12.    Limitations section 4.4

There is a bias described in methods about those who died during the follow-up. Could the authors discuss in some way if there is a high or low mortality rate in >64 years old population that could influence non-participation before and dropout after initiation of the study. As a guide to the reader, authors can provide the life expectancy in Rwanda, which is around 69 years. Maybe, for future studies it would be interesting to change the age groups and decrease last age-group age cut to 60 instead of 64. This change would allow you to have more N in that age-group after loss of follow-up.

Reviewer 2 Report

The manuscript by Nisingizwe et al reports a large national study of anti-HCV screening mostly in adults followed by detection of viraemia determining eligibility for DAA treatment. It shows low yield of viraemia, high success rate of completed treatment and high rates of drop outs.

In introduction, the authors should explain what they mean by ‘HCV cascade’. SVR12 should be spelled out.

M&M. In study context the authors should indicate the cost to the involved population or indicate full cost cover by MoH (issue included only in discussion). The age range and criteria for eligibility should be clearly explained.

In testing section, it is unclear whether, in the public venues cited, only whole blood was collected from which source (venous, capillary) and what volume on what anticoagulant. The main screening assay was SD rapid test. Was testing performed on site or all collected samples were sent to specialised testing labs? It is indicated that these sites were proficient in ELISA testing. Were samples reactive with the RDT re-tested by ELISA and if so which test? Were samples refrigerated for transport and, if not, how long samples were kept at environmental temperature, presumed above 20C? Is the ‘confirmatory’ test mentioned RTD reactive plus  ELISA antibody test reactive or was HCV RNA testing considered confirmatory? The test cascade leading to molecular testing should be clarified (SD RDT to ELISA to RNA?). How is an anti-HCV positive test defined? SD RDT + ELISA reactive?

It appears that HCV RNA testing was performed from a sample different from the initial sample. How was it insured that both samples were taken from the same individual?

The uncertainties regarding the sample collection/ testing should be clarified in a figure reflecting the algorithm used.

There seem to be two levels of confirmation: one at antibody testing level and one after HCV RNA testing. To avoid confusion, the RNA positive samples should be called ‘viraemic’ and eligible for DAA treatment.

The molecular test(s) used in this study should be identified. Was the same test used in all 13 testing labs? And if several tests were used what was the LOD for each assay?

Results

SES should be spelled out and the four categories defined in terms of annual income.

It is unclear why age distribution is limited to < and > 44y. A proper age distribution of the population should be given even if analyses stratify <44 together.

Figure 1. The meaning of anti-HCV diagnosed is unclear. Does it mean ‘confirmed’ anti-HCV positive? i.e. SD RDT and ELISA reactive?

Section 3.3; It is unclear what ‘screening positive’ means? Is it SD RDT reactive or ELISA confirmed positive. Only the latter should be used.

Table 2. As presented, this table is not informative. The actual number of ‘antibody confirmed positive’ and prevalence in each category should be given, excluding negatives. For HCV RNA results, the number and percentage actually tested and % positive should be given.

In Tables 3 and 4, P values should be added.

Section 3.5; should refer to Table 5, not 2.

Section 3.6 mentions base viral load as a factor correlated with treatment failure. Methods to determine viral load should be described in the M&M section. A supplementary figure showing viral load distribution should be constructed and presented.

Discussion

The first paragraph indicates 2.8% of the population carrying detectable HCV RNA. This number is erroneous since it is obtained by dividing the  25,031 samples RNA positive by the total study population of 860,801. In fact, the 25,031 is obtained from only 90% of tested seropositive samples. The denominator should therefore be 774,721 and the prevalence 3.2%.

These numbers are critically important and should be reported in a separate section or table in the result section.

In this regard, the low percentage of RNA positive antibody confirmed positive samples (57%) is unusual and needs to be discussed. In an article of 2003, only 47% of confirmed anti-HCV positive blood donor samples carried HCV RNA and this high prevalence of spontaneously recovered infections was speculated to be related to genotype 2. A highly responsive immune system in SSA populations was an alternative explanation. It would be interesting and highly relevant for the authors to determine the HCV genotype prevalent in Rwanda (J Virol 2003;77:7914-23).

Section 4.1; it is incorrect to mention that HCV is transmitted in fashion similar to HIV since HCV, contrary to HIV, is not sexually transmitted. However, both viruses are transmitted by blood direct contact.

Regarding HBV co-infection in SSA, it should be pointed out that HBV infection occurs early in life, mostly horizontally while HCV infection occur many years later through totally different routes.

Round 2

Reviewer 1 Report

Although some of my previous requests have been omitted, especially those about to improve the data analysis, I will continue to propose minor changes to improve the text.

Abstract:

1)      As a suggestion. Phrase (Lines 21-22): “Of those who confirmed positive, 52% initiated treatment, and 72% of those completed treatment and returned for assessment 12 weeks afterward, and 88% were cured”; can be reformulated as: “Of those who confirmed positive, 52% initiated treatment and 72% of them completed treatment and returned for assessment 12 weeks afterward. Curation rate was 88%.” 

Introduction:

1)      Please note that reference 5 is from 2017 and the text says, “based on 2019 WHO estimates”. Please correct. (Line 38)

2)      Please correct nomenclature in lines 54-55: $ is missing in UD1,200. However, maybe is more appropriate to write from 1,200 to 60 US$.

Material and Methods:

1)      This is a suggestion, but point 2.5 (Line 132) talks about three stages. Maybe points 2.6, 2.7 and 2.8 should be 2.5.1; 2.5.2 and 2.5.3. That change would make point 2.9 to convert in 2.6. (Renumber all sections).

2)      Line 145: Please change Staring April 2017 by Starting April 2017

3)      Figure 1: In figure 1 (second square) the authors write “Patient informed… and sings informed consent”. However, in section 2.13. “Informed consent status” the authors talk about verbal consent. Please correct as corresponds. Was the document read to them and, after, signed or was it read and they approved verbally?

4)      Line 189: Reference 32 doesn’t provide any insights of the way in which SES categories were created. Please, provide other reference or explain in the text.

Results

1)      Lines 231-232: “About half were on the third category of SES”. Would it be possible that missing are separated from category 3 as in table 2 as they are in tables 3 and 4?

2)      Line 233: The author states: “Fewer than 1% did not have health insurance”.  However, in the table 2 the percentage among the total sample that did not have insurance was 18.8%. Please correct as proceed.

3)      This is only a suggestion, but in line 235 it is said: “1.7% had been previously diagnosed with liver disease”. That data do not appear in the Table 2. It is no mandatory that all data appears in the table, but it would be great to see the category on it.

4)      Line 236: “3.4% had a family history of HCV”. In Table 2. % of those with Viral Hepatitis family history was 2.8%. Please correct.

5)      Additionally, please change in Table 2 “Family History of Viral Hepatitis” by “Family History of Viral Hepatitis C” or “Family History of HCV”, the type C is missing. The same for Table 3, 5, and 6.

6)      Lines 305-307: “ Finally, patients with a family history of viral 305 hepatitis (OR: 0.14, 95%CI: 0.08 to 0.24), who ever been operated (OR: 0.39, 95% CI: 0.27 to 306 0.56) and those who were fully or partially vaccinated (OR: 0.14, 95% CI: 0.11 to 0.16) were 307 more likely to undergo their SVR test.” I don’t know were this data come. Please check the ORs as they do not correspond to the table 6. If you are not going to present the model, it is preferable to not present the results so the reader is not confused.

Discussion

1)      Line 315: Please write “Low- and Middle-income Countries (LMICs)” the first time it appears in the text instead of LMIC directly for those who are not familiar with the term.

2)      Is there any plan to reduce the loss of follow-up at this point that can be provided in the discussion that could lead to future studies?

3)      May be the lower effectiveness of the drugs be related to a disruption in the intake of the drug (incorrect daily intake)? Please say something about checking this point, inform the patients about the adherence to treatments relevance and so improve the results.

Please remember that the following points are unresolved from the previous review:

·         Table 4. OR calculated for Hepatitis B Ag results is 15 (too high) in the unadjusted model, and 109.4 in the adjusted model. Does the researchers check for interactions or confounding factors in the final model to explain these results? Could the authors suggest and explanation for these results? Which characteristics has missing group for this variable to obtain such OR values?

·         Section 4.1. Please delete the remark of hypertension as a risk factor to have more HCV. Hypertension and diabetes can be a result of the infection or a result of other risk factors/intrinsic characteristics of patients. In the case of cirrhosis, it is most likely a consequence of HCV infection. Please review all comments made in the manuscript and instead of saying “hypertension or other comorbidity was found to be a risk factor for HCV” say “hypertension or other comorbidity was associated to the presence/diagnosis of HCV”.

Reviewer 2 Report

The revised manuscript is improved but there are still confusion and a need for tightening up before it might be suitable for publication.

In addition to mentioning out-of-pocket payment, the authors should indicate the actual cost in USD.

Table 1 remains in percentages tabulated vertically when it is only interesting and informative if numbers and percentages are given horizontally. This table should be totally reworked out.

Point 3.3 The authors did not clearly answer the question. They considered the result of the RDT at face value which was an error as was clearly demonstrated by several articles showing that specificity of RDTs is very low despite claims by manufacturers and 50-80% of reactive results are false positive. As a result, since there was no systematic serologic confirmation, only HCV RNA positive samples can be considered confirmed as active infection.

It is also confusing to say that capillary blood was used at testing labs when it requires patient’s presence. Was capillary blood used only in patients attending the testing labs or was capillary blood collected in recruitment sites and an anticoagulated vial was transported to testing lab for RDT testing? Ultimately how many samples were tested on venous blood and capillary blood? Is the STD performance comparable with each of these sample sources?

Point 3.4 Now the authors indicate that samples were tested with ELISA at the testing lab but above they say this was not done and only SD RDT results were used. Totally confusing. This need to be sorted out properly.

1.5  Again, more confusion. Now the authors say that samples were tested with either STD or ELISA, not in sequence. How many samples were tested with anti-HCV ELISA? Which test? And how many were tested with STD on capillary blood? What was the respective reactive rate of each type of test and the percentage of the reactive found HCV RNA positive? Such comparison would provide some information regarding the specificity of the RDT.

In the discussion, despite the reply from the authors, it is misleading to indicate that HCV is sexually transmitted, particularly in Africa where male homosexuality, particularly with the rough intercourse described for the rare cases of sexual transmission were reported. This part of the statement should be deleted.
